
# Numerical Bifurcation Methods applied to Climate Models: Analysis beyond Simulation

Henk A. Dijkstra[1,2]

[1]Institute for Marine and Atmospheric research Utrecht, Department of Physics, Utrecht University, Utrecht, the Netherlands
[2]Center for Complex Systems Studies, Utrecht University, Utrecht, the Netherlands
*Correspondence to:* Henk Dijkstra <h.a.dijkstra@uu.nl>

**Abstract.** In this special issue contribution, I provide a personal view on the role of bifurcation analysis of climate models in the development of a theory of climate system variability. The state-of-the-art of the methodology is shortly outlined and the main part of the paper deals with examples of what has been done and what has been learned. In addressing these issues, I will discuss the role of a hierarchy of climate models, concentrate on results for spatially extended (stochastic) models (having
many degrees of freedom) and evaluate the importance of these results for a theory of climate system variability.

## 1   Introduction

The climate system, comprised of the atmosphere, ocean, cryosphere, land and biosphere components, displays variability on a broad range of temporal and spatial scales. Much information on this variability has become available from observations (both instrumental and proxy) over the last decades. Through these observations, many specific phenomena of variability have been
identified, such as the interannual time scale El Niño-Southern Oscillation (ENSO) in the equatorial Pacific (Philander, 1990) and the millennial time scale Dansgaard-Oeschger cycles (Clement and Peterson, 2008).

In classical meteorology, the weather is defined as the variability on a few days time scale and climate is the 'average weather' where usually an averaging time of 30 years is taken. However, such a concept of climate is not very useful as the climate system displays variability over many time scales. Hence, in modern climate dynamics an often used concept is that of
the climate system variability which includes the weather and also variability in the ocean, land, biosphere and ice components. Much of this variability in the climate system is intrinsic (or internal), indicating that it would even exist if the insolation from the Sun would be constant. Intrinsic variability arises through instabilities, in most cases associated with positive feedback processes. There is also variability through radiative forcing variations associated with the diurnal and seasonal cycle and variations of the Earth's orbit (Milankovitch forcing). If we don't consider human activities to be part of the climate system,
then the changes in atmospheric composition due to anthropogenic emissions are also considered as a forcing. The same can be done with lithospheric processes such that volcanic activity is also a forcing component.

Natural climate variability is then all variability due to natural processes (both intrinsic and forced) and anthropogenic climate change is only that part due to human activities. To reliably project future climate change, a thorough knowledge of the natural variability is required. At the moment, there appear to be two different paradigms of natural climate variability





(Fig. 1). One is the classical 'background and peaks' framework (Mitchell, 1976), where the peaks are associated with specific phenomena (Ghil, 2002) such as ENSO (Fig. 1a). Another view is that variability constitutes a continuum of fluctuations with scaling behavior and hence 'peaks' are irrelevant (Lovejoy and Schertzer, 2013). In this paradigm, several different scaling regimes exist up to multi-millennial time scales, e.g., the weather, macro-weather and climate regimes (Fig. 1b). The

'background and peaks' paradigm has clearly a problem with the background signal (which processes cause this variability?) and also regarding the amplitude of the variability on longer than millennial time scales. On the other hand, the scaling paradigm is quite abstract and lacks a clear connection to physical processes beyond the weather regime (Franzke and coauthors, 2019). Both paradigms are also limited to temporal variability and do not address spatial patterns associated with the climate system variability.

To understand the results of the observations, i.e., to relate them to elementary well-established physical principles, the observations themselves are in most cases not enough and models are needed. Fortunately, a hierarchy of such models, from conceptual ones (capturing only a few elementary processes or scales) to global climate models (which are multi-scale and multi-process representations), is available. Traditionally, climate system modeling is seen as an initial value problem. The model equations are integrated in time from a specific initial condition (or an ensemble of them) and then the transient behavior

is analysed. A subsequent statistical analysis is performed on the results using in general uni- or multivariate statistical methods. Often, parameters in the model are varied to study the sensitivity of the results to physical processes (associated with the parameters) and to determine mechanisms of specific phenomena from the statistical analyses.

Changes in parameters can lead to qualitatively different behavior, for example oscillatory behavior appears or transitions occur. When relatively strong changes occur under small changes of a parameter, critical conditions associated with so-called

tipping behavior may have been crossed.

In particular regarding issues of qualitative changes in model behavior once parameters are varied, there is a complementary methodology available from dynamical systems theory, which is targeted to directly compute the asymptotic (long-time) states (attractors) of the model. In the most simple autonomous models (steady forcing), these attractors are fixed points and periodic orbits. Non-autonomous models are studied through pullback attractor analysis (Ghil et al., 2008). There is methodology

available (e.g. from ergodic theory) to study the decay of correlations between different observables to the attractor which can be applied also to stochastic, non-autonomous models (Chekroun et al., 2011).

A canonical problem of transition behavior in fluid dynamics is the flow between two concentric cylinders of which only the inner cylinder rotates with an angular frequency Ω, the Taylor-Couette flow (Koschmieder, 1993). An overview of the regimes of flow behavior and dynamical systems methodology that can be applied is presented in Fig. 2. Here, the rotation rate

of the inner cylinder Ω is used as the main parameter which is changed. In the case of small Ω, numerical bifurcation theory can be applied to study the steady states of these models and to determine mechanisms of transition through instabilities. The transient behavior, as shown in time series in Fig. 2, can also be visualized in phase/state space where time is implicit (displaying trajectories). Once Ω is increased, a collective interaction between the different instabilities can lead to variability which cannot be understood as a single bifurcation. Basically, here the realm of complex systems science is entered, which





deals with emergent properties due to such collective interactions. For large $\Omega$, eventually a turbulent regime is reached where (actually surprisingly) multiple large-scale statistical steady states can be observed (Huisman et al., 2014).

The main issue addressed in this paper is hence whether such a dynamical systems analysis of models of (parts of) the climate system is useful to understand the variability of this system. In section 2, the model hierarchy is sketched and a short overview

will be given of the basic techniques focussing on the application to large-dimensional dynamical systems generated from discretized (stochastic) partial differential equations. In section 3, I will discuss results of studies where dynamical systems analysis has been performed on spatially extended climate models, focussing on what has been done so far and what has been learned. This is followed by section 4 where an outlook is given for the role of dynamical systems analysis in developing an overarching theory of climate variability.

## 2 Methodology

### 2.1 Model hierarchy

In Dijkstra (2013), I suggested to classify climate models according to the two traits 'scales' and 'processes' (Fig. 3). Here the trait 'scales' refers to both spatial and temporal scales as there exists a relation between both: on smaller spatial scales usually faster processes take place. 'Processes' refers to either physical, chemical or biological processes taking place in the different

climate subsystems. Models with a limited number of processes and scales are usually referred to as conceptual climate models. Examples are box models of the ocean circulation (Stommel, 1961) and models of glacial-interglacial cycles (Crucifix, 2012), all formulated by small-dimensional systems of ordinary differential equations. Limiting the number of processes, scales can be added by discretizing the governing partial differential equations spatially up to three dimensions. A higher spatial resolution and inclusion of more processes will give models located in the right upper part of the diagram, so-called Earth System Models

(ESMs). Between the conceptual models and ESMs are so-called intermediate complexity models which are spatially extended (described by partial differential equations) but with fewer scales and/or processes (Fig. 3).

Any spatially extended climate model consists of a set of conservation laws, which are formulated as a set of coupled partial differential equations, that can be written in general form as (Griffies, 2004)

$$\mathcal{M}_\lambda \frac{\partial \mathbf{u}}{\partial t} = \mathcal{L}_\lambda \mathbf{u} + \mathcal{N}_\lambda(\mathbf{u}) + \mathcal{F}_\lambda(\mathbf{u}), \tag{1}$$

where $\mathcal{L}$, $\mathcal{M}$ are linear operators, $\mathcal{N}$ is a nonlinear operator, $\mathbf{u}$ is the state vector, $\mathcal{F}$ contains the forcing of the system and $\lambda$ indicates the dependence of the operators on parameters. Appropriate boundary and initial conditions have to be added to this set of equations for a well-posed problem.

When Eq. (1) is discretized, eventually a set of differential equations with algebraic constraints arises, which can be written as

$$M_\lambda \frac{d\mathbf{x}}{dt} = L_\lambda \mathbf{x} + N_\lambda(\mathbf{x}) + F_\lambda(\mathbf{x}), \tag{2}$$



where $\mathbf{x} \in \mathbb{R}^n$ is the state vector, $n$ its dimension, $M_\lambda$ is a (often singular) matrix of which every zero row is associated with an algebraic constraint, $L_\lambda$ is the discretized version of $\mathcal{L}_\lambda$, and $F_\lambda$ and $N_\lambda$ are the finite-dimensional versions of the forcing and the nonlinear operator, respectively.

When noise is added, the evolution of the flow can generally be described by a stochastic differential-algebraic equation of the form

$$M_\lambda \, \mathrm{d}\mathbf{X}_t = \mathbf{f}_\lambda(\mathbf{X}_t) \, \mathrm{d}t + \mathbf{g}_\lambda(\mathbf{X}_t) \circ \mathrm{d}\mathbf{W}_t, \tag{3}$$

where $\mathbf{X}_t$ is the stochastic state vector, $\mathbf{f}_\lambda(\mathbf{X}_t)$ contains linear, nonlinear processes and the forcing, $\mathbf{W}_t \in \mathbb{R}^{n_w}$ is a vector of $n_w$-independent standard Brownian motions (Gardiner, 2009), and $\mathbf{g}_\lambda(\mathbf{X}_t) \in \mathbb{R}^{n \times n_w}$. The $\circ$ symbol indicates that the Stratonovich interpretation is used as the noise is believed to represent unresolved processes.

## 2.2 Continuation methods

These methods form part of the numerical bifurcation analysis toolbox; here we restrict to a single parameter $\lambda$. Finding steady states of the system (2) versus $\lambda$ amounts to solving

$$\mathbf{f}_\lambda(\mathbf{x}) = L_\lambda \mathbf{x} + N_\lambda(\mathbf{x}) + F_\lambda(\mathbf{x}) = 0. \tag{4}$$

The idea of pseudo-arclength continuation (Keller, 1977; Seydel, 1994) is to parametrize branches of solutions $\Gamma(s) \equiv (\mathbf{x}(s), \lambda(s))$ with an arclength parameter $s$ (or an approximation of it, thus the term 'pseudo') and choose $s$ as the continuation parameter. An additional equation is obtained by approximating the normalization condition of the tangent $\dot{\Gamma}(s) = (\dot{\mathbf{x}}(s), \dot{\lambda}(s))$ to the branch $\Gamma(s)$, where the dot refers to the derivative with respect to $s$, with $|\dot{\Gamma}|^2 = 1$. More precisely, for a given solution $(\mathbf{x}_0, \lambda_0)$, the next solution $(\mathbf{x}, \lambda)$ is required to satisfy the constraint

$$\dot{\mathbf{x}}_0^T(\mathbf{x} - \mathbf{x_0}) + \dot{\lambda}_0(\lambda - \lambda_0) - \Delta s = 0, \tag{5}$$

where $\dot{\Gamma}_0 = (\dot{\mathbf{x}}_0, \dot{\lambda}_0)$ is the normalized direction vector of the solution family $\Gamma(s)$ at $(\mathbf{x}_0, \lambda_0)$ and $\Delta s$ is an appropriately small step size. Equation (5) stipulates that the projection of $(\mathbf{x}, \lambda) - (\mathbf{x}_0, \lambda_0)$ onto $(\dot{\mathbf{x}}_0, \dot{\lambda}_0)$ has the value $\Delta s$. Efficient solution methods for high-dimensional systems of the form (4-5) are presented in De Niet et al. (2007) and Thies et al. (2009).

Suppose that the deterministic part of Eq. (3) has a stable fixed point $\mathbf{x}_\lambda^*$ for a given range of parameter values. Then linearization of (3) around the deterministic steady state yields (Kuehn, 2012)

$$M_\lambda \, \mathrm{d}\mathbf{X}_t = A_\lambda(\mathbf{x}_\lambda^*)\mathbf{X}_t \, \mathrm{d}t + B_\lambda(\mathbf{x}_\lambda^*) \circ \mathrm{d}\mathbf{W}_t, \tag{6}$$

where $A_\lambda(\mathbf{x}) \equiv (D_\mathbf{x}\mathbf{f}_\lambda)(\mathbf{x})$ is the Jacobian matrix and $B_\lambda(\mathbf{x}) = \mathbf{g}_\lambda(\mathbf{x})$.

In the special case that $M_\lambda$ is a non-singular matrix, the equation (6) can be rewritten (dropping the arguments and subscripts on the matrices) in Itô form as,

$$\mathrm{d}\mathbf{X}_t = M^{-1}A \, \mathbf{X}_t \, \mathrm{d}t + M^{-1}B \, \mathrm{d}\mathbf{W}_t, \tag{7}$$





which represents an $n$-dimensional Ornstein-Uhlenbeck process. The corresponding stationary covariance matrix $C$ is then determined from the generalized Lyapunov equation

$$ACM^T + MCA^T + BB^T = 0. \tag{8}$$

The Gaussian probability density function at the fixed point can then be computed directly from $C$. When $M$ is singular, special methods have been devised to cope with the singular part (Baars et al., 2017); also in this case, a generalized Lyapunov equation determines the covariance matrix $C$. Efficient solution methods for high-dimensional versions of (8) are presented in Baars et al. (2017).

## 3 Main issues

Numerical bifurcation methodology has been mostly applied to dynamical systems with small $n$, typically $n < 10$, resulting from conceptual climate models. However, I will focus here solely on results of studies using intermediate complexity models with typically $n = 10^4 - 10^5$, because then also spatial information on the bifurcation behavior is obtained.

From elementary bifurcation theory (Guckenheimer and Holmes, 1990) it is known that only four bifurcations can occur when a single parameter is varied: the saddle-node bifurcation, the transcritical bifurcation, the pitchfork bifurcation and the Hopf bifurcation. Because the transcritical bifurcation (solution needed for all values of the parameter) and the pitchfork bifurcation (reflection symmetry needed) require special conditions, the only generic cases are the saddle-node and the Hopf bifurcations. Of these, the saddle-node only occurs in pairs (because of boundedness of solutions) and hence one often refers to a back-to-back saddle-node bifurcation. While the saddle-node is a critical transition, the Hopf bifurcation is not as in the latter case nearby solutions exist on both sides of the bifurcation (Kuehn, 2011).

The back-to-back saddle-node bifurcation structure is canonical for tipping points, which we will discuss in section 3.1 below. Although the dynamical system is high-dimensional, the behavior of the system can be dominated by only a few (even only one) positive feedbacks and hence transitions occur in a low-dimensional space. The Hopf bifurcation is canonical for the occurrence of spontaneous oscillatory behavior associated with one eigenmode of the linearized dynamical system, which is often referred as the leading mode. A Hopf bifurcation needs the presence of both positive and negative feedbacks; when only a few dominate the dynamical behavior these can be found in high-dimensional systems as discussed in section 3.2. In models where a sequence of Hopf bifurcations occurs, the resulting behavior can in general no longer be described using low-dimensional dynamics. In this case, collective interactions occur and this cannot be captured in a single bifurcation and associated pattern. This case will be discussed in section 3.3 below.

### 3.1 Tipping points

An overview of possible tipping elements in the Earth's system was given in Lenton et al. (2008) and a recent (Steffen et al., 2018) overview is shown in Fig 4a. Several of these transitions are thought to be associated with the existence of a multiple equilibrium regime associated with a back-to-back saddle node structure, in particular the collapse of the Atlantic Meridional Overturning Circulation (AMOC) and that of Marine Ice Sheets (MIS).





For a back-to-back saddle node bifurcation there are two transition scenarios possible, called (i) bifurcation tipping and (ii) noise-induced tipping (Ditlevsen and Johnsen, 2010). In case (i) the parameter crosses a value at one of the saddle node bifurcations and in case (ii) a finite amplitude perturbation in the state vector causes a transition (even for a fixed value of the parameter). In the non-autonomous case also rate-induced tipping (Ashwin et al., 2012) is possible. For both cases (i) and (ii),

it is crucial to determine the extend of the multiple equilibrium regime (Fig 4b): this has been investigated in detail in spatially extended models for the following problems.

- AMOC: In a substantial number of papers, the bifurcation diagrams for both spatially two- and three-dimensional (intermediate complexity) ocean-only models of the AMOC has been determined (see chapter 6 in Dijkstra (2005)). The
most advanced result is for a global ocean model coupled to an energy balance atmospheric model (Toom et al., 2012), capturing ocean-atmosphere feedbacks, where also a back-to-back saddle-node structure was found.

  Numerical bifurcation analyses provided the basis for the stability indicator $\Sigma = M_{ov}^s - M_{ov}^n$ of the multiple equilibrium regime of the AMOC (Huisman et al., 2010). Here, $M_{ov}$ is the MOC induced freshwater transport and the superscripts $n$ and $s$ indicate the northern and southern boundaries of the Atlantic, respectively. Although the central idea was al-
ready formulated in Rahmstorf (1996) and de Vries and Weber (2005), only bifurcation analysis of high-dimensional discretized ocean models provided more rigorous support for the use of this indicator. Although this stability indicator has its problems (Gent, 2018) and needs to be extended in a coupled ocean-atmosphere context, it is much used to investigate the stability of the AMOC (Hawkins et al., 2011).

  For a spatially two-dimensional ocean-only model, the covariance matrices $C$ were determined from solving a Lyapunov
equation (8) in Baars et al. (2017) for the case of noise in the freshwater forcing. While here it served only to test the new Lyapunov equation solver (RAILS), the methodology was extended recently to compute (noise-induced) transition probabilities of the AMOC and to relate that probability to the stability indicator $\Sigma$ (Castellana et al., 2019). Such transitions are thought to be involved in the Dansgaard-Oeschger (DO) events (Ditlevsen and Johnsen, 2010).

- MIS: The explicit computation of the bifurcation structure of a spatially one-dimensional marine ice-sheet model (with
a moving grounding line) has been carried out only recently (Mulder et al., 2018). Here also a back-to-back saddle node structure is found, of course compatible with the many initial value problem studies of this model (Schoof, 2007). The gravitational effect of a marine ice sheet on sea level has a stabilizing influence on the ice sheet as seen through a shift in the bifurcation diagram. While this was found in many different model studies (Gomez et al., 2010), the precise mechanism could be deduced from the bifurcation diagram shift (Mulder et al., 2018).

For a stochastic MIS model, the covariance matrices $C$ were determined for each stable state in Mulder et al. (2018). Typical noise in the accumulation leads to grounding line variations in the order of 1000 m, while for sea level noise this is about 100 m. In the multiple equilibrium regime, the study of the transition probabilities indicated that for both noise types, it is more likely to jump from a large ice sheet state to a small ice sheet state than vice versa.




In high-dimensional climate models, also so-called edge states or Melancholy states have been computed, for example in a coupled atmospheric sea-ice model investigating ice covered/ice free multi-stability (Lucarini and Bódai, 2017). This edge state is a saddle embedded in the boundary between the two basins of attraction of the stable climate states.

## 3.2 Patterns of SST variability

It is remarkable that on interannual-to-multidecadal time scales the variability in sea surface temperature is organized in large-scale patterns (Fig. 5). These patterns have been detected by using multi-variate analysis on long data sets, such as the HadISST, for example through principle component analysis where the patterns are then contained in the empirical orthogonal functions (EOFs). Well known and much studied patterns are those of the El Niño-Southern Oscillation (ENSO), the Atlantic Multi-decadal Oscillation (Enfield et al., 2001) and the Pacific Decadal Oscillation (Mantua et al., 1997) as shown in Fig. 5.

Numerical bifurcation analysis has been applied to several spatially extended models, in particular of ENSO, PDO and the AMO.

   – ENSO: The cornerstone intermediate complexity model is the Zebiak-Cane (ZC) model of which the behavior has been extensively analysed (Zebiak and Cane, 1987). Numerical bifurcation analysis of different versions of the ZC model were performed (see chapter 7 in Dijkstra (2005)) and in each of them a Hopf bifurcation occurs once the coupling
strength between the equatorial ocean and atmosphere crosses a critical value. The period of the leading mode is in the interannual range and determined by basin modes, just as in the recharge-discharge oscillator model (Jin, 1997). The spatial pattern of the leading mode is localized into the cold tongue region of the mean (steady) state and shares many similarities with the first EOF from observations.

   – AMO: One of the intermediate complexity models which has been used is a spatially three-dimensional model of the
North Atlantic (see chapter 8 in Dijkstra (2013)) The mean (steady) state is generated by a horizontal atmospheric surface buoyancy field which drives a meridional overturning circulation in the ocean model. Numerical bifurcation analysis of this model has shown that the background state destabilizes through a multi-decadal leading mode and hence a Hopf bifurcation occurs. The time scale of the leading mode can be linked to the basin crossing time of temperature anomalies and its spatial pattern shares many features with the observed pattern (Kushnir, 1994) when a representation of the
continents is considered. The variability can be easily excited through noise in the heat flux, even when the leading mode is decaying in the deterministic case (see chapter 8 in Dijkstra (2013)).

   – PDO: The same bifurcation analysis as for the AMO was applied to a model of two ocean basins which are connected by a Southern Ocean (von der Heydt and Dijkstra, 2007). It was found that the PDO cannot be related to a single mode of variability which arises through a Hopf bifurcation (as for ENSO and the AMO). The key here is that destabilization
of the mean (steady) flow can only occur when there is sinking (by the MOC) in the northern part of the basin. Such sinking is absent in the North Pacific for the present day climate. Indeed, modern views of the PDO indicate that several different mechanisms are likely important for the existence of PDO variability (Newman et al., 2016).


My interpretation of these results is that several of these SST patterns (but not all) appear through a normal mode which destabilizes the mean state through positive feedbacks; the presence of negative feedbacks cause the oscillatory behavior. In this case, the associated Hopf bifurcation (of a spatially extended model) provides both the dominant time scale of variability and its spatial pattern. The elegant structure of leading modes in ocean models and the ZC model was presented in Dijkstra

(2016). The key why normal modes can be dominant in this variability may be that the nonlinearity in these models is rather weak (it involves advection of heat/salt and not of momentum) and hence the mean state is not modified (rectified) much due to the nonlinear interactions (contrary to the processes shown in the next section).

### 3.3  Collective interactions and emergent behavior

In the previous two subsections climate system variability phenomena were attributed to low-order dynamics. However, there

are many phenomena which are intrinsically caused by the collective interaction of multiple instabilities. Clearly, the role of numerical bifurcation theory becomes quite limited in determining the behavior of these (in general) chaotic dynamical systems; I briefly describe below two examples.

- Ocean western boundary current variability (WBC): In pure wind-driven barotropic shallow-water models, the bifurcation structure consists of an imperfect pitchfork bifurcation followed by several Hopf bifurcations containing two type

of modes: Rossby modes and so-called gyre modes. These gyre modes are eventually responsible for homoclinic orbits which lead to ultra-low frequency behavior (see chapter 5 in Dijkstra (2005)). When another layer is added, baroclinic instabilities lead to a range of normal modes (Simonnet et al., 2003) which all destabilize and hence the eventual emergent behavior is induced by their collective interactions. Such behavior can lead to low-frequency behavior in the statistical steady state which has been referred to as the turbulent oscillator (Berloff et al., 2007).

- Midlatitude atmospheric variability (MAV): Analysis of barotropic spatially extended models of midlatitude atmospheric flows has shown that there are many (highly) unstable equilibria (Legras and Ghil, 1983; Crommelin, 2003). Also here the resulting variability occurs through a collective interaction and low-frequency variability emerges in the statistical equilibrium state (Crommelin et al., 2004). Approaches to understand the dynamics on the attractor have been proposed through transfer operator methods (Tantet et al., 2015).

The cases briefly described above are examples of strongly nonlinear systems, where the nonlinearities occur in the momentum advection and where the mean state is strongly modified through rectification. Of course, there are many more examples of such geophysical systems, in particular on time scales up to interannual both in the ocean (internal waves) and the atmosphere (weather).

### 4  Discussion and Outlook

In this paper, I have given a short overview of results of studies where continuation methods were applied to spatially extended climate models. My interpretation of these results is that there are climate variability phenomena that can be attributed to





low-order behavior; only one or a few number of spatial patterns are involved associated with dominant feedbacks. Several of these studies have also shown that also successive instability behavior can occur. This leads to a collective interaction between patterns that is eventually responsible for emerging variability in climate models. A summary of the different phenomena based on this distinction is provided in Fig. 6, with low-order phenomena (ENSO, AMO), emergent phenomena due to collective

interactions (WBC, AMV) and tipping behavior (AMOC/DO, MIS). With this information, I now come back to the two paradigms of climate variability, as mentioned in the introduction.

This first challenge I see is to better understand processes behind the background variability which is 'red noise like' in Mitchell (1976) and encompasses all variability in Lovejoy et al. (2013); Rypdal and Rypdal (2014). Franzke and coauthors (2019) describe the different ways (multifractal cascading processes, state-dependent noise, etc.) how scaling behavior can

appear in time series. To connect scaling behavior to clear physical processes, one idea would be to identify at each time scale range the 'slow' passive component in the climate system and the 'fast' forcing it receives. For example, for SST variability the Hasselmann (1976) model of an ocean mixed layer forced with a rapidly fluctuating atmospheric heat flux identifies such components, leading to a red noise background. However, when another variable is considered, such as sea surface height, then the background model is an ocean thermocline layer forced with noisy wind-stress forcing, leading to a Correlated Additive

Multiplicative (CAM) noise model (Sardeshmukh and Sura, 2009; Castellana et al., 2018). In the 'climate regime' of Lovejoy et al. (2013), an appropriate model could be a marine ice sheet forced with rapidly fluctuating accumulation noise or sea level noise (Mulder et al., 2018), leading also a CAM noise. This would give power law spectral behavior (in the proxy record), being indeed very different from the picture of Mitchell (1976). The slope changes of the different regimes as in Lovejoy et al. (2013) could then maybe be related to a change in slow component in the climate system determining this background signal.

Once the physics of this background are clear, the next challenge is to attribute spatial patterns which rise above it to specific instabilities. Several spatial patterns of SST variability are robust over the model hierarchy. I would interpret this to indicate that spatial patterns, such as ENSO and the AMO, are due to a single mode destabilization of the background induced by dominant large-scale feedbacks. These spatial patterns can already be captured in detail in intermediate complexity models, such as the ZC model for ENSO. Capturing the temporal variability involves representation of small-scale processes (noise)

and possibly non-normal growth (Penland and Sardeshmukh, 1995; Farrell and Ioannou, 1996; Tziperman et al., 2008) and one may need more detailed models than intermediate complexity models. Another case where a low-order explanation may be appropriate is centennial variability. The time scale here arises by buoyancy anomalies which propagate over the AMOC loop (also called overturning modes and loop oscillations). These can be excited in this model by applying noise, e.g. in the freshwater flux or in the heat flux (Dijkstra and von der Heydt, 2017).

Other spatial patterns (such as the MAV and WBC variability) arise through a collective interaction of instabilities and hence can only be captured in detail through models high in the hierarchy (representing a multitude of scales). This holds for example for the path variability of the Kuroshio Current, where it is known that the interactions of the barotropic instabilities of the current and the (baroclinic) mesoscale eddy field are important (Qiu and Chen, 2005). Such collective interactions may eventually also be needed to explain the PDO. Furthermore, analysis of high-resolution (near-eddy resolving) ocean model has

indicated that a new type of multidecadal variability emerges through a collective interaction, the Southern Ocean Mode. A





Lorenz type energy analysis has indicated (Jüling et al., 2019) that eddy mean-flow interaction is crucial for the existence of this type of variability. Also in this case, there is no single (normal) mode of variability which determines the dominant time and spatial pattern.

Apart from the internal variability introduced by single normal (oscillatory) modes and collective phenomena, also clear
large-scale tipping phenomena (in the sense of critical transitions) can affect climate variability. The canonical behavior is a back-to-back saddle node bifurcation appearing generically in conceptual models. It was shown here that for models of the AMOC and MIS, indeed such bifurcation behavior is found in high-dimensional models. Transition behavior hence may occur when critical conditions are crossed or through noise in the multiple equilibrium regime. A third challenge I see is to show that such transitions remain robust once small-scale processes are included; work in this direction has been initiated (Lucarini and
Bódai, 2017).

All of the results of continuation methods described above were obtained under stationary forcing and for many in the field this seems disjoint from the real climate system, which is obviously forced by a non-stationary insolation component (on diurnal, seasonal and orbital time scales). For the present-day climate system, there is also the non-stationary anthropogenic component of climate change. A fourth challenge is to understand the relevance of these diurnal and seasonal non-stationary
periodic components in natural internal variability on longer time scales. While one may argue that they are irrelevant and are averaged out, few detailed results are available. Probably only on interannual time scales, there can be an interaction between the seasonal cycle and internal variability, for example with the ENSO mode (due to nonlinear resonances). On very large time scales, however, certainly the non-stationary orbital forcing is crucial for the observed variability such as glacial cycles. The modification of natural variability under climate forcing is of course also a challenging issue.

Has the end point been reached of the models for which bifurcation analysis can be applied? Since starting with this endeavour in the early 1990's, I was repeatedly asked this question. When we showed results for spatially two-dimensional ocean models, we were asked if we could do this for three-dimensional models. When we did, the question was on the application to ocean-atmosphere models. Although there are certainly still interesting details to be investigated in the ocean-only context, I think the main challenge with these models is to develop theory for internal variability in the geological past (Zachos et al.,
2001). In the last few years my group has turned to develop techniques to incorporate sea ice and land ice and to be able to change the geometry of the ocean basin during the continuation (Mulder et al., 2017). With a carbon cycle model still to be implemented, I think that the resulting methodology will be suited to tackle what processes determined the background states in the past, which variability can possibly be attributed to low-order dynamics. It will also be possible to investigate explicit bifurcation behavior arising from carbon-cycle feedbacks.

*Acknowledgements.* The author thanks his 'partner-in-crime' Dr. ir. F.W. Wubs (University of Groningen, the Netherlands) for the (now ∼ 25 year) long and very fruitful (still very active) collaboration regarding the application of numerical bifurcation analysis in high-dimensional stochastic dynamical systems arising from climate models. I also thank the many contributions from the PhDs and Postdocs who have worked in our joint projects, in particular the current ones Erik Mulder, Sven Baars and Daniele Castellana. This work was sponsored by



the Netherlands Earth System Science Center (NESSC) financially supported by the Ministry of Education, Culture and Science (OCW), grant no. 024.002.001. I also thank Andrew Keane and Bernd Krauskopf (University of Auckland, NZ) for organizing a session on SIAM-Dynamical Systems (May 2019) on Climate Feedbacks where the ideas in this paper could be presented.





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

(a)

(b)

**Figure 1.** (a) The 'background and peaks' paradigm as an artist's view of climate system variability (figure slightly modified from Mitchell (1976), see Ghil (2002) for details). (b) A composite temperature spectrum as in Lovejoy et al. (2013) (see their Fig. 2 for details) to illustrate the 'scaling' paradigm.



**Figure 2.** Sketch of dynamical systems concepts and approaches for the Taylor-Couette flow (as modified from Abraham and Shaw (1992)). Time series, trajectories and the geometrical view of attractors are sketched. Transition behavior at small values of $\Omega$ can be addressed by bifurcation theory, for large values of $\Omega$ it can be tackled using ergodic theory.





Figure 3. Organization of climate models according to the two model traits: number of processes and number of scales (Dijkstra, 2013).


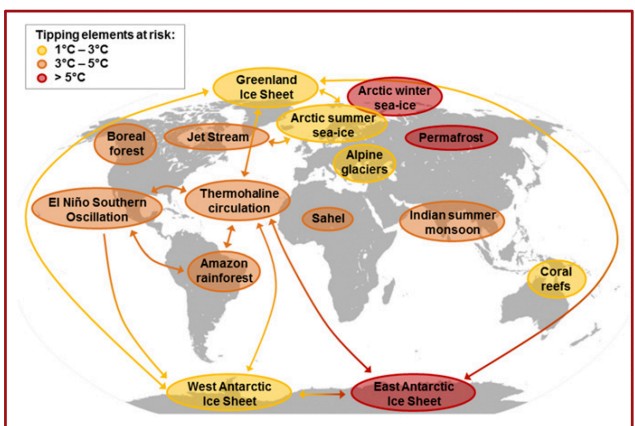

(a)

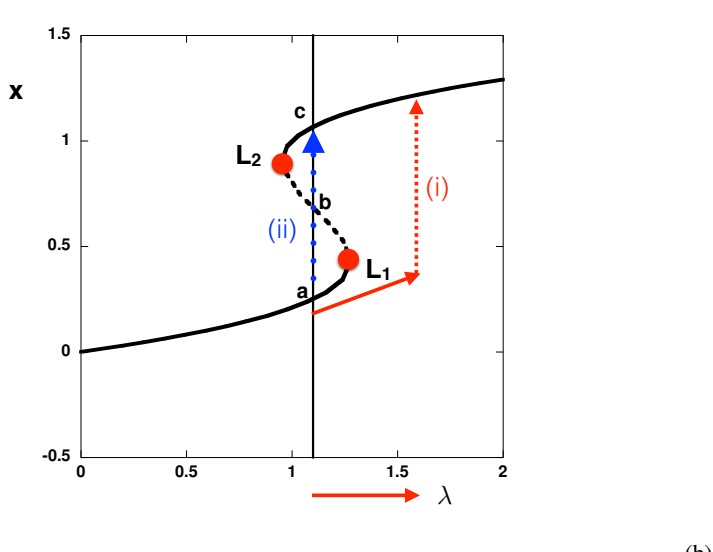

(b)

**Figure 4.** (a) Tipping elements in the Earth system, after Lenton et al. (2008) but the figure is from Steffen et al. (2018). (b) The canonical bifurcation diagram with the back-to-back saddle-node indicating two stable states (**a** and **c**) and an unstable state (**b**). Bifurcation tipping occurs when the parameter $\lambda$ crosses the value at $L_1$ or $L_2$. Noise induced tipping (e.g. from state a to state c) can occur through a perturbation in the state vector (for fixed $\lambda$).

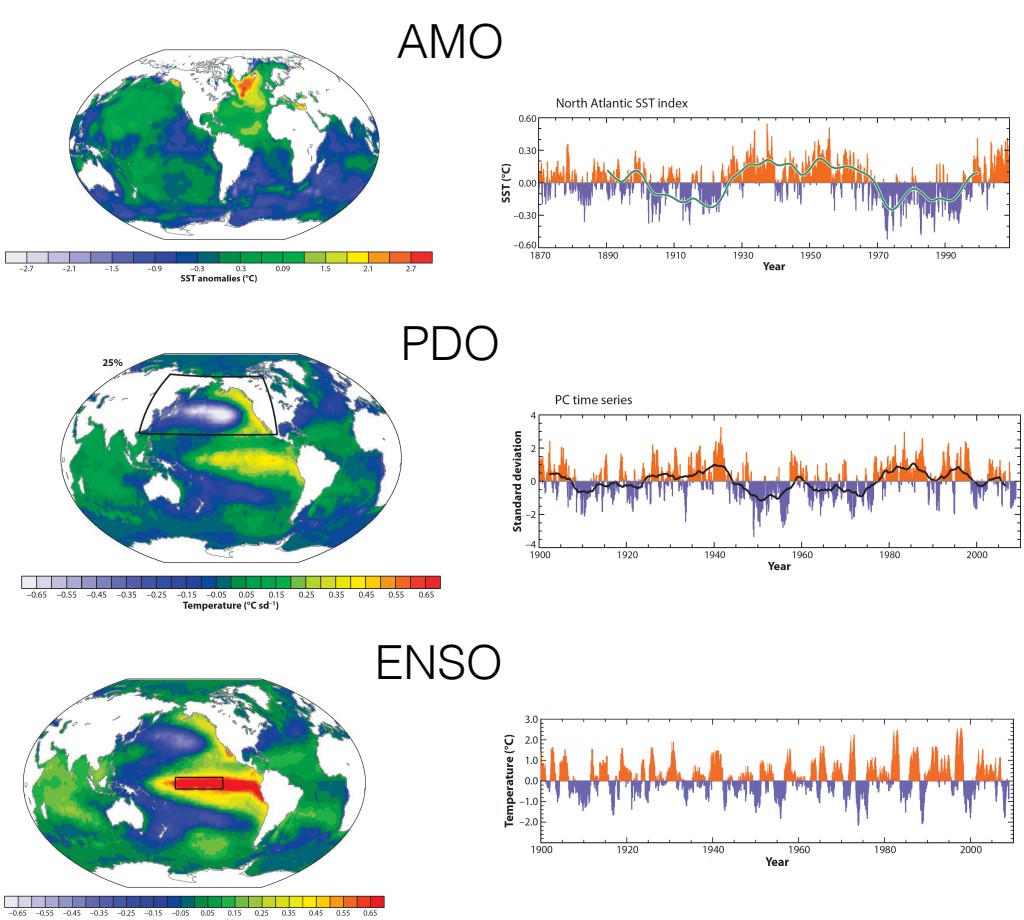

**Figure 5.** Overview of patterns of climate variability (AMO, PDO and ENSO) as determined in Deser et al. (2010) with accompanying time series.

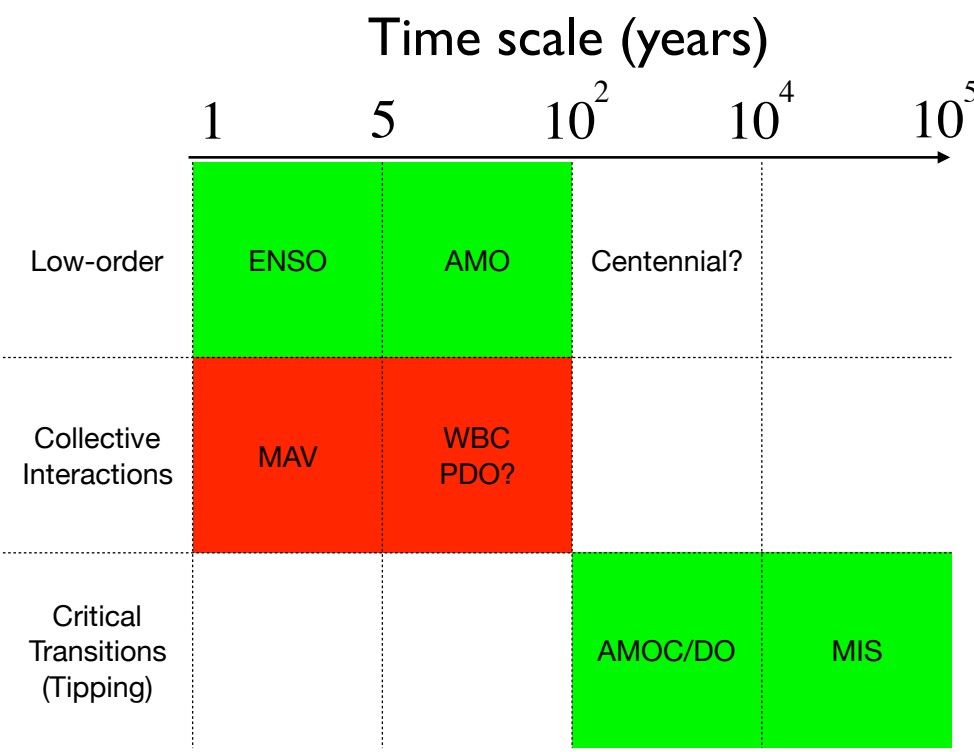

**Figure 6.** Summary of what has been learned from dynamical systems analysis of spatially extended climate models, based on the distinction of low-order phenomena, emerging phenomena through collective interactions and critical transitions. The 'hope' is that mechanisms of the phenomena in the green boxes can be determined from numerical bifurcation analysis of intermediate complexity climate models.