# Peer review of "Numerical Bifurcation Methods applied to Climate Models: Analysis beyond Simulation"

_Nonlinear Processes in Geophysics, 2019_

## Short Comment (SC1) · 21 Jun 2019

"All of the results of continuation methods described above were obtained under stationary forcing and for many in the field this seems disjoint from the real climate system, which is obviously forced by a non-stationary insolation component (on diurnal, seasonal and orbital time scales). "

Are tidal forcing factors considered on orbital time scales? According to Munk and Wunsch, tidal factors are a factor in overturning circulation.

1. Munk, W. & Wunsch, C. Abyssal recipes II: energetics of tidal and wind mixing. Deep Sea Research Part I: Oceanographic Research Papers 45, 1977–2010 (1998).

---

## Referee Comment (RC1) · Anonymous Referee #1 · 22 Jun 2019

Recommendation: Minor revisions

1) Page 2, line 3: I do not think that Lovejoy and Schertzer really believe that peaks in spectra are irrelevant. The diurnal and annual cycles are definitely of importance. I think they want to emphasize that the background spectrum is also important.

2) Page 2, lines 24-26: I understand the first part of the sentence but not the second part. This sentence should be rewritten.

3) Page 4, line 9: Also Ito noise can represent unresolved processes. The point of using Stratonovich noise is that those unresolved processes are serially correlated; whereas Ito noise would assume that the noise is serially uncorrelated.

4) A brief explanation what x and X stand for would and whether there is any real

difference be helpful.

5) AMOC, MIS, etc should be defined at first usage.

6) Page 8, line 14: "two type" -> "two types"

7) Page 9, line 17: "also a CAM" -> "also to a CAM"
* * *

---

## Referee Comment (RC2) · Anonymous Referee #2 · 30 Jun 2019

The author, one of the main developers of this approach, reviews and discuss what has been attained with the use of bifurcation methods of climate models of increasing complexity. Although personal in style, I find the overview quite complete and illustrative of the state of the field, providing adequate background and references, and identifying remaining challenges.

I recommend publication of the paper. I just point out some minor points that the author can correct in the final version:

- In Eq. (6), the state vector $X_t$ should be replaced by the difference with the deterministic steady state, say $Y_t$, defined as $Y_t = X_t - x^*_\lambda$

-Define MOC before it appears first (in page 6). What has been defined in page 5 is

AMOC. The relationship between the two acronyms would be quite evident for most readers, but could be confusing for the ones less familiar with this type of circulation.

- Page 7: principle component -> principal component

- Page 5: '... only four bifurcations can occur GENERICALLY when a single ...'

- I am not happy with the nomenclature of 'critical transitions'. The reason is that the expanding tendency in environmental applications is to use it in the sense of 'abrupt', 'discontinuous', which is exactly the contrary of the much older use of the word in several fields of physics, qualifying phase transitions. Perhaps a less ambiguous name would be 'dangerous transition' (Thompson et al. (1994) Phys. Rev. E 49, 1019 ; Int. J. Biff. Chaos (2011) 21, 399). I understand that the use of 'critical' is now prominent in many fields of science and I do not request the author to correct all the paper with this respect. But there is one place, the sentence '... the saddle-node is a critical transition, the Hopf bifurcation is not ...' in page 5, in which changing 'critical' to 'discontinuous' or some related word would make it less strange to readers with some particular backgrounds.

---

## Author Response (AR1)

**Response to Anonymous Referee RC 1**
I thank the referee for the careful reading and the useful comments and will adapt the manuscript accordingly. Below is a point by point reply with the referee's comments in bold font, my reply in italic font and the changes in manuscript in normal font.

1. Comment of the referee:
**Page 2, line 3: I do not think that Lovejoy and Schertzer really believe that peaks in spectra are irrelevant. The diurnal and annual cycles are definitely of importance. I think they want to emphasize that the background spectrum is also important.**

Author's reply:
*Agreed.*

Changes in text:
Paragraph will be adapted to more accurately describe the Lovejoy and Schertzer view.

2. Comment of the referee:
**Page 2, lines 24-26: I understand the first part of the sentence but not the second part. This sentence should be rewritten.**

Author's reply:
*This is the motivation for continuation methods which was probably too much compactly formulated.*

Changes in text:
Sentences will be split to clarify the main point.

3. Comment of the referee:
**Page 4, line 9: Also Ito noise can represent unresolved processes. The point of using Stratonovich noise is that those unresolved processes are serially correlated; whereas Ito noise would assume that the noise is serially uncorrelated.**

Author's reply:
*Agreed.*

Changes in text:
This aspect will be mentioned in the revised text.

4. Comment of the referee:
**A brief explanation what x and X stand for would and whether there is any real difference be helpful.**

Author's reply:
Agreed.

Changes in text:
The interpretation of both symbols will be better described.

5. Comment of the referee:
**AMOC, MIS, etc should be defined at first usage.**

Author's reply:
*Agreed.*

Changes in text:
The introduction and explanation of the acronyms will be checked and corrected.

6. Comment of the referee:
**Page 8, line 14: "two type" -> "two types"**

Author's reply:
*Agreed.*

Changes in text:
Will be corrected.

7. Comment of the referee:
**Page 9, line 17: "also a CAM" -> "also to a CAM"**

Author's reply:
*Agreed.*

Changes in text:
Will be corrected.

**Response to Anonymous Referee RC 2**
I thank the referee for the careful reading and the useful comments and will adapt the manuscript accordingly. Below is a point by point reply with the referee's comments in bold font, my reply in italic font and the changes in manuscript in normal font.

1. Comment of the referee:
**In Eq. (6), the state vector X_t should be replaced by the difference with the deterministic steady state, say Y_t, defined as Y_t = X_t - xˆ*_nlambda**

Author's reply:
*That is indeed better.*

Changes in text:
Formula will be adapted.

2. Comment of the referee:
**Define MOC before it appears first (in page 6). What has been defined in page 5 is AMOC. The relationship between the two acronyms would be quite evident for most readers, but could be confusing for the ones less familiar with this type of circulation.**

Author's reply:
*This was indeed unclear.*

Changes in text:
The explanation of the acronyms will be checked and corrected.

3. Comment of the referee:
**Page 7: principle component -> principal component**

Author's reply:
*Agreed.*

Changes in text:
Text will be corrected.

4. Comment of the referee:
**Page 5: '... only four bifurcations can occur GENERICALLY when a single ...'**

Author's reply:
*Agreed.*

Changes in text:
Will be corrected.

5. Comment of the referee:
**I am not happy with the nomenclature of 'critical transitions'. The reason is that the expanding tendency in environmental applications is to use it in the sense of 'abrupt', 'discontinuous', which is exactly the contrary of the much older use of the word in several fields of physics, qualifying phase transitions. Perhaps a less ambiguous name would be 'dangerous transition' (Thompson et al. (1994) Phys. Rev. E 49, 1019 ; Int. J. Biff. Chaos (2011) 21, 399). I understand that the use of 'critical' is now prominent in many fields of science and I do not request the author to correct all the paper with this respect. But there is one place, the sentence '... the saddle-node is a critical transition, the Hopf bifurcation is not ...' in page 5, in which changing 'critical' to**

**'discontinuous' or some related word would make it less strange to readers with some particular backgrounds.**

Author's reply:
*Agreed.*

Changes in text:
Text will be adapted.

**Response to** Paul Pukite (SC1)
I thank Dr. Pukite for his interesting short comment on the paper.  Below is the  comment in bold font, my reply in italic font and the changes in manuscript in normal font.

1. Comment of the referee:
**"All of the results of continuation methods described above were obtained under stationary forcing and for many in the field this seems disjoint from the real climate system, which is obviously forced by a non-stationary insolation component (on diurnal, seasonal and orbital time scales). "**

**Are tidal forcing factors considered on orbital time scales? According to Munk and Wunsch, tidal factors are a factor in overturning circulation.**
**1. Munk, W. &Wunsch, C. Abyssal recipes II: energetics of tidal and wind mixing. Deep Sea Research Part I: Oceanographic Research Papers 45, 1977–2010 (1998).**

Author's reply:
*Tidal factors are certainly important the maintain the mean state ocean circulation on long time scales, but they are usually not considered when looking at orbital variations, where changes in this mean state are considered. Effectively, they are represented at a high aggregate level by the vertical mixing coefficients in the ocean model component.*

Changes in text:
No changes in the text needed.

[revised manuscript text omitted]

Time Series

Trajectories in Phase space

Attractors in Phase space

Physical space

Bifurcation Theory

Ergodic theory

**Figure 2.** Sketch of dynamical systems concepts and approaches for the Taylor-Couette flow (as modified from Abraham and Shaw (1992)). Time series, trajectories and the geometrical view of attractors are sketched. Transition behavior at small values of $\Omega$ can be addressed by bifurcation theory, for large values of $\Omega$ it can be tackled using ergodic theory.

[Figure]

**Figure 3.** Organization of climate models according to the two model traits: number of processes and number of scales (Dijkstra, 2013).

[Figure]

**Figure 4.**  The canonical bifurcation diagram with the back-to-back saddle-node indicating two stable states (**a** and **c**) and an unstable state (**b**). Bifurcation tipping occurs when the parameter $\lambda$ crosses the value at $L_1$ or $L_2$. Noise induced tipping (e.g. from state a to state c) can occur through a perturbation in the state vector (for fixed $\lambda$).

[Figure]

**Figure 5.** Overview of patterns of climate variability (AMO, PDO and ENSO) as determined in Deser et al. (2010) with accompanying time series.

[Figure]

**Figure 6.** Summary of what has been learned from dynamical systems analysis of spatially extended climate models, based on the distinction of low-order phenomena,  emergent phenomena through collective interactions and critical transitions. The 'hope' is that mechanisms of the phenomena in the green boxes can be determined from numerical bifurcation analysis of intermediate complexity climate models.